# Carbonate Buffer Mixture Alleviates Subacute Rumen Acidosis Induced by Long-Term High-Concentrate Feeding in Dairy Goats by Regulating Rumen Microbiota

**DOI:** 10.3390/microorganisms13040945

**Published:** 2025-04-19

**Authors:** Guyue Fan, Nier Su, Yuhong He, Chongshan Yuan, Caijun Zhao, Xiaoyu Hu, Yunhe Fu, Naisheng Zhang

**Affiliations:** Department of Clinical Veterinary Medicine, College of Veterinary Medicine, Jilin University, Changchun 130062, China; fanguyue1994@163.com (G.F.); suniersunier@163.com (N.S.); heyh24@mails.jlu.edu.cn (Y.H.); yuancs2021@163.com (C.Y.); zhaocj2001@sina.com (C.Z.); hxiaoyu@yeah.net (X.H.)

**Keywords:** rumen pH, LPS, inflammation, barrier permeability

## Abstract

This study aimed to elucidate the therapeutic mechanisms of carbonate buffer mixture (CBM) in mitigating subacute rumen acidosis (SARA) by examining its effects on rumen pH, systemic inflammation, and rumen microbiota in a dairy goat model. Using a controlled experimental design, SARA was induced through 8-week high-concentrate diet feeding (70% concentrate, 30% forage), followed by 2-day CBM treatment. Comprehensive analyses included rumen pH monitoring, serum inflammatory marker quantification (IL-1β, TNF-α) by ELISA, rumen barrier integrity assessment through tight junction proteins (TJs) ZO-1, Occludin, and Claudin-3 by western blot analysis, and 16S rRNA sequencing of rumen microbiota. The results demonstrated that CBM administration rapidly elevated depressed rumen pH within 6 h post-treatment while concurrently reducing circulating LPS levels. The analysis of rumen 16S rRNA showed that CBM significantly increased the rumen microbial diversity and abundance of SARA dairy goats. Butyric acid generation groups such as *Rikenellaceae_RC9_gut_group*, *NK4A214_group,* and *Prevotellaceae UCG-001* were selectively enriched, and corresponding functional predictions showed that the butyric acid synthesis pathway (PICRUSt2) was enhanced. These findings suggest that CBM has a multidimensional therapeutic effect by simultaneously correcting rumen acidosis, alleviating systemic inflammation, and restoring microbial balance through pH-dependent and pH-independent mechanisms, providing a scientifically validated nutritional strategy for SARA management in intensive ruminant production systems.

## 1. Introduction

The escalating global demand for dairy products has driven the widespread adoption of high-concentrate diets (HCDs) in modern ruminant production systems. Scientific evidence demonstrates that HCDs significantly enhance propionate-dominated rumen fermentation [1], which serves as the primary substrate for hepatic gluconeogenesis. This metabolic adaptation can significantly improve the feed conversion rate (FCR) compared to traditional feed-based feeding regimens [2]. However, when ruminants consume HCDs, due to insufficient effective physical fiber provided by the low percentage of roughage in the diet, the chewing frequency decreases ruminating insufficiently, and salivary secretion decreases. This leads to a decrease in the amount of bicarbonate and dihydrogen phosphate entering the rumen and a decrease in the ability of the rumen to neutralize organic acids. Short-chain fatty acids (SCFAs) are absorbed by the rumen wall at a lower rate than produced, which causes the pH of the rumen to decrease [3,4,5]. Meanwhile, a large amount of rapidly digestible starch (RDS) was degraded by rumen microorganisms to produce lactic acid [6], and an elevated proportion of lactic acid converted to butyric acid by lactic acid-utilizing bacteria increased [7]. However, when lactic acid production exceeds the metabolic capacity of lactic acid-utilizing bacteria, the rumen pH rapidly decreases and disturbs the rumen microbiota, inducing subacute rumen acidosis (SARA) [8]. The impaired rumen environment characteristic of SARA significantly diminishes fiber fermentation efficiency while inducing structural damage to the rumen epithelium. These pathological changes lead to marked reductions in the feed conversion ratio (FCR) and predispose dairy ruminants to a spectrum of metabolic disturbances and infectious complications, particularly mastitis [9] and endometritis [10], that impair the health of dairy animals.

Studies have shown that rumen microorganisms contain the largest proportion of bacteria and are highly sensitive to changes in rumen fluid pH [11,12]. When SARA occurs, acid-intolerant rumen bacteria die off in large numbers, causing changes in rumen bacterial community abundance and diversity, resulting in damage to rumen tissues and a reduced FCR [13]. It was also found, in our previous study, that the abundance and diversity of bacterial communities in the rumen fluid of dairy cows were significantly reduced when SARA was established by high-concentrate feeding [9]. When rumen pH decreased, the abundance of Firmicutes and Actinobacteria increased, and Proteobacteria and Bacteroidetes decreased [14]. Some studies have found that, after SARA occurs in sheep, due to the death and lysis of Gram-negative bacteria such as Bacteroides and the release of endotoxin (LPS), the relative abundance of Bacteroidetes decreases, while the concentration of rumen LPS increases significantly [15]. Free LPS in the rumen enters the blood circulation through the damaged rumen barrier and activates the body’s immune response [11]. Research has demonstrated that high-dose yeast culture supplementation (SCFPb-2X) exerts beneficial effects in SARA-affected dairy cows by selectively promoting cellulolytic bacteria and lactate-utilizing microorganisms while stabilizing ruminal pH and reducing free LPS concentrations [16]. In a complementary investigation, Petri et al. revealed that the combined administration of autolyzed yeast (AY) and phytogenic (PHY) through concentrate mixtures significantly modulated rumen microbiota, notably reducing the abundance of Gram-negative *Succiniclasticum* spp. while enhancing populations of lactate-utilizing *Selenomonas* spp. [17].

Building upon these established findings, the present study aims to investigate whether CBM could regulate rumen microbiota while regulating rumen pH parameters. Specifically, we seek to elucidate the potential dual mechanisms through which CBM may mitigate SARA-associated inflammation via direct pH stabilization and through selective microbial population shifts that could reduce LPS-producing bacterial taxa while promoting beneficial commensals. This study will provide important insights into CBM as a therapeutic mechanism for SARA in ruminants and may provide synergistic benefits through physicochemical and microbial pathways.

## 2. Materials and Methods

### 2.1. Reagents

The reagents including Na_2_CO_3_, NaHCO_3_, NaCl, MgSO_4_, CaCl_2_, and KCl were obtained from Tianli Chemical Reagent Co., Ltd. (Tianjin, China). The LPS test kit (cat.YX-121618G) was obtained from Horseshoe Crab Reagent Manufactory Co., Ltd. (Xiamen, China). Tumor necrosis factor (TNF)-α (cat.430915) and interleukin (IL)-1β (cat.432615) enzyme-linked immunosorbent assay (ELISA) kits were obtained from BioLegend (San Diego, CA, USA). Specific antibodies including beta-actin (β-actin) (#AF7018), Zonula occludens-1 (ZO-1) (#AF5145), Occludin (#DF7504), and Claudin-3 (#AF0129) were obtained from Affinity Biosciences (Cincinnati, OH, USA).

### 2.2. Animals and Experimental Protocol

Thirty dairy goats (2–3 years, average weight of 55 kg) were obtained from a farm in Changchun, Jilin Province, China, with no diseased animals and no history of antibiotic therapy. All diets met the daily nutritional requirements needed for lactation of dairy goats and were randomly allocated into three groups (n = 10); the Ctrl group received a basal diet containing 30% concentrate (dry matter basis) with a 7:3 forage-to-concentrate ratio; the SARA model group underwent gradual adaptation where the concentrate proportion increased from 30% to 70% over 21 days to induce SARA, maintaining the final 7:3 ratio for 8 weeks; and the CBM treatment group received oral administration of a CBM mixture, which was mainly composed of Na_2_CO_3_ (25 g), NaHCO_3_ (200 g), NaCl (50 g), MgSO_4_ (25 g), CaCl_2_ (2.5 g), and KCl (10 g) dissolved in 5 L of water. In terms of the dosing of SARA-affected dairy goats, each dairy goat was dosed once a day for a total of 2 days.

The experimental diets comprised a series of nutritionally balanced formulations with increasing concentration levels from 30% to 70% of DM. The formulations systematically replaced alpha hay (decreasing from 56% to 26% DM) with higher proportions of corn meal (32–56% DM) and soybean meal (8–14% DM), while maintaining constant inclusion rates of limestone (2% DM), salt (1% DM), and a vitamin/mineral premix (1% DM). This premix provided comprehensive micronutrient supplementation, containing 450 mg nicotinic acid, 600 mg Mn, 950 mg Zn, 430 mg Fe, 650 mg Cu, 30 mg Se, 45 mg I, 20 mg Co, 800 mg vitamin E, 45,000 IU vitamin D, and 120,000 IU vitamin A per kilogram. The nutritional profiles demonstrated progressive increases in metabolic energy (8.19–18.34 MJ/kg DM) and starch content (17.12–37.95% DM), while maintaining consistent crude protein levels (15% DM) across all diets. Corresponding reductions in fiber fractions were observed, with neutral detergent fiber decreasing from 52% to 25% DM and acid detergent fiber from 35% to 15% DM.

### 2.3. Rumen pH and LPS Testing

After 8 weeks of HCD feeding, a gastric catheter tube was inserted into the esophagus and delivered to the rumen using a hand-pressure gastric suction device, and the rumen contents were extracted, and the pH was detected by a pH meter after filtration. After determining that the SARA model had been successfully established, the rumen fluid was collected by rumenocentesis using nonpyrogenic needles and syringes. The samples were centrifuged at 10,000× *g* for 45 min at 4 °C, and the supernatant was aspirated and passed through a 0.22 mm LPS-free filter to harvest the supernatant for the LPS assay.

### 2.4. Histopathologic Analysis

All rumen and gut tissues were fixed in 4% paraformaldehyde, embedded in paraffin, and sectioned at 5 μm. After dewaxing and hydration, sections were stained with hematoxylin and eosin (H&E). Histopathological evaluation was performed under a microscope (Olympus Corporation, Tokyo, Japan) to assess the severity of epithelial injury (from absent to mild) and the extent of inflammatory cells in the filtrate (from none or rare to transmural). A semi-quantitative scoring system adapted from [9] was applied to grade the severity of each parameter.

### 2.5. Comprehensive Metabolic Panel, Blood Biochemical Analysis, and LPS Tests of Serum

Blood samples were collected from the jugular vein of goats using a standardized protocol. The first sample was drawn into a vacuum blood collection tube containing ethylenediaminetetraacetic acid (EDTA) as an anticoagulant for complete blood count (CBC) analysis. The second sample was collected in a sterile, pyrogen-free microcentrifuge tube and subsequently processed by centrifugation at 1000× *g* for 10 min at 4 °C. The resulting serum supernatant was carefully aliquoted for subsequent biochemical parameter quantification and LPS detection assays.

### 2.6. Analysis of Milk Composition

The dairy goat teat area was sterilized using 75% ethanol, and the first three handfuls of milk were discarded to minimize contamination. Approximately 50 mL of milk was then aseptically collected into a sterile tube containing potassium dichromate as a preservative. Milk composition analysis was performed using a Milkoscan™ FT1 infrared analyzer (FOSS Analytical A/S, Hillerød, Denmark) to determine fat, protein, lactose, milk density, non-fat milk, and ash content.

### 2.7. ELISA

The collected rumen tissue was mixed with PBS in a 1:9 ratio to produce a 10% tissue homogenate. Subsequently, the sample was centrifuged at 12,000× *g* for 10 min at 4 °C, and the supernatant was collected for detection. Blood samples are centrifuged under the same conditions to collect the upper serum. The pro-inflammatory cytokines TNF-α and IL-1β were measured in tissue supernatants and serum samples using commercially available ELISA kits, following the manufacturer’s protocols. Briefly, 100 μL of standards or diluted samples (1:5 in assay buffer) were added to antibody-precoated wells and incubated for 2 h at 37 °C. After washing five times with 0.5% TBST, 100 μL of biotinylated detection antibody was added (1 h, 37 °C), followed by HRP-streptavidin (30 min) and TMB substrate (15 min). The reaction was stopped with H_2_SO_4_, and absorbance was measured at 450 nm (reference 570 nm).

### 2.8. Western Blot

After 0.03 g of rumen tissue was weighed, 270 μL of RIPA lysate was added, followed by grinding and centrifugation at 12,000× *g* and 4 °C for 10 min to collect the supernatant. Protein concentration was determined using the BCA protein assay kit, and total protein was extracted for later use. The target protein was isolated using 12% SDS-PAGE, wet transformed into a “sandwich”, and sealed in 5% buttermilk at room temperature for 3 h. After overnight incubation at 4 °C with specific primary antibodies, the PVDF membranes were washed three times (5 min each) with 0.5% TBST buffer. The antibody was then incubated at room temperature for 2 h, the PVDF membrane was cleaned with 0.05% TBST, and the system was imaged. Goat anti-rabbit antibodies or goat anti-mouse antibodies were incubated at room temperature for 2 h, PVDF membranes were cleaned with 0.5% TBST, and imaging was performed in the ECL detection system.

### 2.9. Rumen Fluid DNA Extraction and Illumina MiSeq Sequencing

Total bacterial DNA extraction Microbial community genomic DNA was extracted from rumen samples of dairy goats using the E.Z.N.A.^®^ soil DNA kit (Omega Bio-tek, Norcross, GA, USA) according to the manufacturer’s instructions. After testing the DNA quality with 2% agarose gel, the DNA concentration and purity were determined using NanoDrop2000 (Thermo Fisher Scientific, Waltham, MA, USA). Using the ABI GeneAmp^®^ 9700 PCR thermal circulator (Applied Biosystems, Foster City, CA, USA), Bacterial 16S rRNA genes V3-V4 were amplified with 338F (5′-ACT CCT ACG GGA GGC AGC AG-3′) and 806R (5′-GGA CTA CHVGGG TWT CTAAT-3′). The amplification conditions of the 16S rRNA gene were as follows: predenaturation at 95 °C for 3 min, 27 cycles (denatured at 95 °C for 30 s, annealed at 55 °C for 30 s, and extended at 72 °C for 30 s),extended for 10 min at 72 °C, and stored at 4 °C. Products of PCR were recovered with 2% agarose gel, purified according to the instructions of the DNA Gel Recovery and Purification Kit (MajorBio, Shanghai, China) and quantified with Qubit 4.0 (Thermo Fisher Scientific). The products were sequenced using the Illumina PE300/PE250 platform (Illumina, San Diego, CA, USA) according to the standard protocol of Majorbio. UPARSE v7.1 was used to perform operational taxonomic unit (OTU) clustering and eliminate chimeras based on 97% similarity of the quality control concatenated sequences. The PERMANOVA nonparametric test was used to analyze the differences in microbial community structure between groups. To characterize intergroup variations in microbial composition, LDA effect size (LEfSe) analysis was employed to detect phylogenetically discriminative features with statistical significance. The functional potential of the microbiota was subsequently predicted through phylogenetic investigation of communities by the reconstruction of unobserved states (PICRUSt2) based on 16S rRNA gene sequencing data.

### 2.10. Statistical Analysis

All data in this study were analyzed and plotted using GraphPad Prism 8.0 (GraphPad Software, San Diego, CA, USA). One-way analysis of variance (ANOVA) was used to analyze the mean difference in the normal distribution data. * *p* ≤ 0.05, ** *p* ≤ 0.01, *** *p* ≤ 0.001, or **** *p* ≤ 0.0001; these experimental results were considered to have significant differences.

## 3. Results

### 3.1. CBM Alleviates SARA Induced by High-Concentration Diet in Dairy Goats

Rumen pH is a common criterion for diagnosing SARA. The results indicate that the rumen fluid pH of dairy goats in feeding with an HCD for 8 weeks was consistently lower than that of the Ctrl group, remaining below 5.8 for three consecutive days and for more than 3 h per day, indicating that the SARA modeling was successful. However, rumen fluid pH returned to the normal range within 24 h after CBM treatment (Figure 1A). In addition, dry matter intake (DMI) was reduced in SARA dairy goats, while treatment with CBM significantly increased DMI in dairy goats during SARA (Figure 1B). The results showed that CBM could significantly alleviate SARA.

### 3.2. The Effects of CBM on CBC in SARA Dairy Goats

Blood samples were taken from three groups of dairy goats for CBC. The results showed that the SARA group had a significantly higher white blood cell count (WBC) compared to the Ctrl group (*p* < 0.001). Specifically, the percentage of neutrophils (Neu%) was significantly increased (*p* < 0.001), while there was no significant change in the percentage of lymphocytes (Lym%). However, in the CBM-treated group, both WBC and Neu% values were significantly decreased compared to the SARA dairy goats (*p* < 0.01; Figure 2A–C). In addition, in the SARA group of dairy goats, the values of red blood cell count (RBC) and hemoglobin (HGB) concentrations were reduced, and the platelet count (PLT) was decreased compared to the ctrl group (*p* < 0.01, *p* < 0.0001, *p* < 0.0001, respectively). However, in the CBM-treated group, treatment with CBM significantly increased the values of RBC and HGB and increased the values of PLT during SARA in dairy goats (*p* < 0.05, *p* < 0.001, *p* < 0.001, respectively; Figure 2D–F).

### 3.3. The Effects of CBM on Blood Biochemical Analysis in SARA Dairy Goats

Blood biochemical analysis results showed that the serum concentration of aspartate aminotransferase (AST), total protein (TP) (both *p* < 0.001), globulin (GLB) (*p* < 0.01), albumin (ALB) (*p* < 0.0001), and the albumin–globulin ratio (A/G) (*p* < 0.0001) were significantly increased in the SARA group compared to the Ctrl group. However, CBM reduced the concentration of AST, TP, GLB, and A/G compared with the SARA group (*p* < 0.01, *p* < 0.05, *p* < 0.05, *p* < 0.01, respectively; Figure 3A–E). In addition, except for the total bile acid (TBA), CBM reduced the increases in serum total bilirubin (TB) and alkaline phosphatase (ALP) levels in the SARA group (*p* < 0.05 and *p* < 0.01, Figure 3F–H).

### 3.4. The Effects of CBM on LPS and the Concentration of Inflammatory Cytokines in SARA Dairy Goats

The LPS content in the rumen and circulation was significantly higher in the SARA group (*p* < 0.001 and *p* < 0.01), but its content was significantly lower after CBM treatment (*p* < 0.001 and *p* < 0.05, Figure 4A,B). The activated immune cells’ pattern recognition receptors (PRRs) can identify pathogen-associated molecular markers (PAMPs), such as LPS. Through the activation of downstream signaling pathways, they release various inflammatory cytokines that participate in inflammatory processes. It is shown that the concentrations of TNF-α and IL-1β in serum were significantly higher in the SARA group compared to the Ctrl group (both *p* < 0.0001). However, their concentrations were significantly lower in the CBM-treated group (both *p* < 0.0001, Figure 4C,D). The results indicate that CBM treatment may reduce SARA-induced systemic inflammation in dairy goats.

### 3.5. The Effects of CBM on Milk Composition in SARA Dairy Goats

When LPS translocases into the peripheral blood circulation and causes metabolic changes in the body, nutrients are redistributed, and a large amount of nutrients are used for the body’s immune response, thus reducing the amount of nutrients going to the mammary glands for the synthesis of milk composition, reducing the quality of the milk. In the SARA group, we found that the content of milk fat and lactose decreased significantly (*p* < 0.001 and *p* < 0.05), but the synthesis rate of milk fat and lactose increased after CBM treatment (both *p* < 0.05; Figure 4E,F). In the SARA model, milk protein and milk density tended to decrease, but their contents did not change after CBM treatment (*p* = 0.9733 and *p* = 0.8017; Figure 4G,H). In addition, non-fat milk, ash, and other milk compositions did not change in the two groups (*p* = 0.9773 and *p* = 0.9522, Figure 4I,J).

### 3.6. The Effects of CBM on Rumen and Gut Barrier During SARA in Dairy Goats

From the pathohistological changes, the rumen epithelium of dairy goats was detached at the occurrence of SARA, and there was a large infiltration of inflammatory cells. However, after treatment with CBM, the inflammatory infiltration in the rumen epithelium of dairy goats was significantly reduced (*p* < 0.05, Figure 5A,B). We further investigated the effect of CBM treatment on rumen barrier permeability. Changes in the rumen epithelial tight junction proteins (TJs) ZO-1, Occludin, and Claudin-3 were detected. SARA was found to cause a significant decline in TJ expression, but CBM could restore this (*p* < 0.0001, *p* < 0.05, *p* < 0.05, respectively; Figure 5C–F). In addition, the gut tissues of dairy goats were also tested, and the results were the same as those of rumen tissues. After CBM treatment, the inflammatory cell infiltration decreased (*p* < 0.01, Figure 6A,B), and the expression of TJs was increased (*p* < 0.05, *p* < 0.05, *p* < 0. 001, respectively; Figure 6C–F). These results suggest that CBM therapy can improve barrier permeability by upregulating rumen and gut TJs.

### 3.7. CBM Treatment Reversed Rumen Microbiota Dysbiosis SARA in Dairy Goat

We further investigated whether CBM regulated rumen microecology while alleviating SARA-induced inflammation. Principal Coordinate Analysis (PCoA) using unweighted UniFrac distances showed a significant separation of the rumen microbiota structure between the control, SARA, and CBM groups (R = 0.6255, *p* = 0.001, Figure 7A). Alpha diversity analysis (ACE, Chao1, and Simpson) showed that estimators of community diversity and richness of rumen microbiota were reduced in the SARA group. However, they were significantly increased after CBM treatment (Figure 7B–D). Furthermore, the four dominant phyla in all samples were Firmicutes, Bacteroidetes, Actinobacteria, and Patescibacteria (Figure 7E), and the relative abundances of Actinobacteria and Cyanobacteria were significantly increased, whereas Firmicutes, Bacteroidetes, Patescibacteria, and other low-abundance phyla including Proteobacteria, Verrucomicrobiota, Chloroflexi, Spirochaetota, and Synergistota were significantly reduced in the SARA group compared to the Ctrl group. However, these changes were reversed after the CBM treatment. In addition, we further analyzed the effects of CBM on community composition at the genus level. The results indicate that *Rikenellaceae_RC9_gut group*, *NK4A214_group*, *norank f_Eubacterium_coprostanoligenes_group*, *Olsenella,* and *Prevotellaceae_UCG-001* were significantly decreased in the SARA group compared with the Ctrl group. However, there was a significant increase in their abundance after CBM treatment (Figure 7F). To further identify the differential genera, we used the linear discriminant analysis (LDA) effect size method (LEfSe). The results showed that the relative abundance of *Rikenellaceae_RC9_gut_group* and *Eubacterium_coprostanoligenes_group* had the highest significance in the Ctrl group, whereas *Christensenellaceae_R-7_group* and *Acetitomaculum* represented the major populations in the SARA group. The major populations of the CBM group consisted of *Candidatus_Saccharimonas*, *NK4A214_group*, *Prevotellaceae_UCG-003,* and *Eubacterium_nodatum_group*. Additionally, the CBM group had a higher number of taxa with differential abundance compared to the SARA group (Figure 7G), indicating that CBM treatment has a positive effect on increasing specific microbiota in the SARA group. These data demonstrate that CBM can regulate the composition of rumen microbiota.

### 3.8. CBM Treatment Altered Rumen Microbiota Function and Phenotype in SARA Dairy Goat

Functional prediction of rumen microbiota under CBM treatment was performed using PICRUSt. The predicted functional profiles were subsequently annotated and analyzed through the Kyoto Encyclopedia of Genes and Genomes (KEGG) database to elucidate potential metabolic pathways. Based on the results, we identified the enzyme, ORTHOLOGY (KO), modules, and pathways that belong to KEGG functional categories were identified. Furthermore, the SARA group dramatically affected the four functional categories compared with the Ctrl group. However, these effects were completely restored by the CBM treatment (Figure 8A–C). Thus, CBM treatment improves rumen microbiota functions involved in inflammatory injury, metabolism, immune response, and pathopoiesia. In addition, based on the BugBase phenotype prediction of the Ctrl group, SARA group, and CBM group, we found that the relative abundance of Stress Tolerant and Anaerobic was significantly decreased in the SARA group compared to the Ctrl group, while the Potentially Pathogenic, Forms Biofilms, and Facultatively Anaerobic were significantly increased. In addition, the proportion of Gram-negative tended to decrease, while the proportion of Gram-positive tended to increase in the SARA group. Interestingly, under the CBM treatment, Anaerobic and Forms Biofilms obviously decreased. In addition, CBM treatment also reduced the Gram-negative and induced the Gram-positive, but there was no significant difference (Figure 8D,E).

## 4. Discussion

The development of SARA in ruminants is predominantly associated with the sustained administration of high-concentrate rations to enhance the FCR. It is characterized by prolonged rumen pH below 5.8 and rumen microbiota disturbance. When SARA occurs in dairy goats, it causes metabolic disorders such as diarrhea, decreased dry matter intake and weight loss, and decreased milk production, resulting in huge economic losses to the dairy industry [12]. The treatment of SARA can be completed in various ways, such as by increasing the proportion of roughage in the diet to promote saliva secretion [18,19,20], adding plant extracts to enhance fermentation, and incorporating probiotics to inhibit lactobacillus. A solution of CBM is a basic solution composed of inorganic compounds such as Na_2_CO_3_, NaHCO_3_, MgSO_4_, NaCl, MgSO_4_, CaCl_2_, and KCl. It can neutralize excess free acid in the rumen, which can quickly relieve the decrease in rumen pH. The aim was to investigate whether CBM could alleviate SARA and possible regulatory mechanisms.

Ruminant SARA was diagnosed when rumen pH remained <5.8 for ≥3 h/day [21]. After 8 weeks of HCD feeding, the SARA model was successfully established and then treated with CBM for 2 days. Results indicate that CBM effectively alleviated SARA-induced rumen pH dysregulation by chemically buffering excess volatile fatty acids (VFAs) and free H+ ions. In contrast, NaHCO_3_ enhanced rumen buffering capacity over time by stimulating salivary secretion and modulating rumen osmotic pressure in SARA-affected goats [22]. The decline in rumen pH promoted the proliferation and lysis of Gram-negative bacteria, releasing substantial amounts of LPS. Upon binding to TLR4, LPS activated the NF-κB/MAPK signaling pathway, triggering the excessive production of pro-inflammatory cytokines (e.g., TNF-α, IL-1β) [23]. Consistent with this mechanism, SARA-affected dairy goats in our study exhibited a marked increase in systemic inflammation. Simultaneously, excessive LPS reduced milk fat percentage in SARA-affected animals by decreasing ruminal acetic acid (a key milk fat precursor) and inhibiting critical mammary lipogenic enzymes, including acetyl-CoA carboxylase and fatty acid synthetase [24]. Furthermore, LPS upregulated matrix metalloproteinases (MMPs) [25], degraded TJs, compromised rumen barrier integrity, translocated to mammary tissue, and ultimately impaired mammary epithelial cell function, leading to metabolic dysfunction, reduced glucose availability, and decreased lactose synthesis [26]. These findings were consistently observed in our experimental results. Notably, CBM treatment effectively mitigated these adverse effects by significantly increasing rumen pH in SARA-affected dairy goats. Previous studies have demonstrated that exogenous additives can modulate rumen microbiota to reduce LPS levels, thereby alleviating SARA [16,17]. Building on this evidence, we hypothesized that CBM could further optimize rumen microbial composition by suppressing Gram-negative bacteria proliferation and consequently limiting LPS production. To test this hypothesis, we investigated the regulatory effects of CBM on rumen microbiota.

Sequencing analysis of 16S rRNA revealed that CBM treatment effectively restored SARA-induced alterations in rumen microbial diversity and abundance. At the phylum level, CBM administration significantly increased Firmicutes abundance, a Gram-positive bacteria phylum, thereby competitively suppressing Gram-negative bacterial proliferation. As the primary nutrient source for gastrointestinal epithelial cells, butyrate levels demonstrate a positive correlation with gastrointestinal permeability [27]. In our study, the enriched Firmicutes phylum was identified as the predominant functional bacterial group responsible for butyrate synthesis, with notable producers including *Lachnospiraceae*, *Ruminococcaceae*, *Eubacteriaceae*, and *Fusobacteriaceae* [28,29,30]. Butyrate has been well documented to possess anti-inflammatory properties, offering protection against gastrointestinal pathogens through the maintenance of bacterial enzyme activity. Simultaneously, the abundances of Bacteroidetes and Verrucomicrobiota were significantly elevated in the CBM-treated group. Bacteroidetes contribute to rumen epithelial energy metabolism through SCFA production, while Verrucomicrobiota enhance barrier function by modulating mucus layer thickness [31,32]. This study further demonstrated that CBM treatment significantly increased Actinobacteria abundance, which exhibited a negative correlation with inflammatory markers (WBC and Neu%). Additionally, CBM administration markedly enhanced the abundance of beneficial microbiota including *Rikenellaceae_RC9_gut_group*, *NK4A214_group*, and *Prevotellaceae_UCG-001*. These microbiota-derived metabolites modulate rumen epithelial cell function, proliferation, and secretory activity through multiple signaling pathways, thereby influencing rumen motility and enhancing barrier integrity [33,34,35]. Furthermore, this microbial–metabolic crosstalk impacts host metabolic homeostasis and ameliorates both local and systemic inflammatory responses.

## 5. Conclusions

This study substantiates our initial hypothesis that CBM exerts therapeutic effects on SARA through integrated physicochemical and microbial regulatory mechanisms. The findings demonstrate CBM’s capacity to address our research objectives by simultaneously modulating rumen pH parameters and microbial community structure, thereby attenuating both local and systemic inflammatory responses. These results advance our understanding of nutritional interventions for SARA by establishing CBM’s dual-action mechanism that reconciles microbial ecology with rumen physiological homeostasis. This evidence supports the potential application of CBM in the treatment of SARA in ruminants and sheds light on its mechanisms.

## Figures and Tables

**Figure 1 microorganisms-13-00945-f001:**
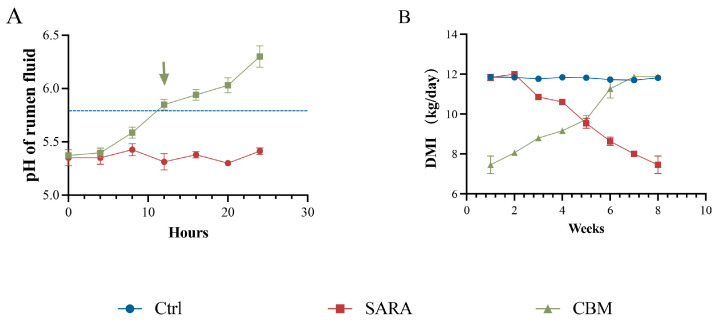
CBM alleviates SARA induced by an HCD in dairy goats. The SARA model was established by feeding dairy goats with an HCD with a ratio of 7:3, and then CBM was used for treatment. (**A**) Dynamics of rumen fluid pH in dairy goats of SARA and CBM groups at the same time of day (n = 3). (**B**) DMI of dairy goats in the Ctrl group, SARA group, and CBM group (n = 3). Data are presented as an XY Plot and means ± SDs.

**Figure 2 microorganisms-13-00945-f002:**
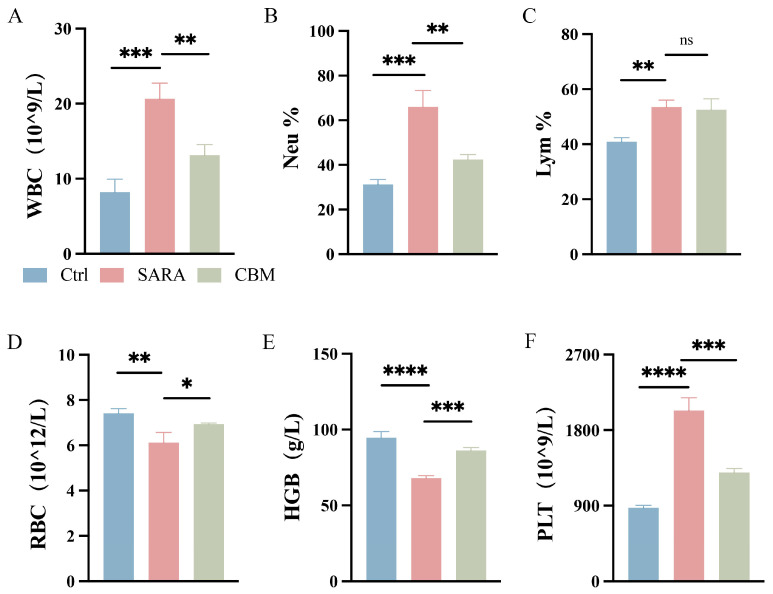
The effects of CBM on Blood-RT in SARA dairy goats. Blood samples of dairy goats in Ctrl group, SARA group, and CBM group were collected for CBC detection. Changes in (**A**) WBC, (**B**) Neu%, (**C**) Lym%, (**D**) RBC, (**E**) HGB, and (**F**) PLT in blood samples of three different groups of dairy goats (n = 3). * *p* < 0.05, ** *p* < 0.01, *** *p* < 0.001, and **** *p* < 0.0001 indicate significant differences, ns, no significance. Data are displayed as means ± SDs, and one-way ANOVA was performed for statistical analysis.

**Figure 3 microorganisms-13-00945-f003:**
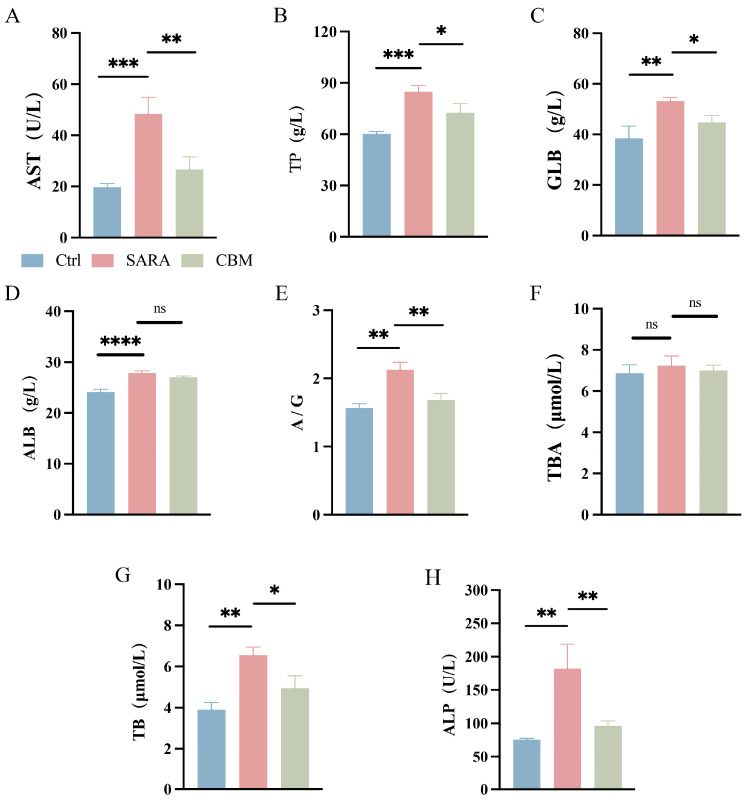
The effects of CBM on blood biochemical analysis in SARA dairy goats. Blood samples of dairy goats in Ctrl group, SARA group, and CBM group were collected for blood biochemical analysis. Changes in (**A**) AST, (**B**) TP, (**C**) GLB, (**D**) ALB, (**E**) A/G, (**F**) TBA, (**G**) TB, and (**H**) ALP in blood samples of three groups of dairy goats (n = 3). * *p* < 0.05, ** *p* < 0.01, *** *p* < 0.001, and **** *p* < 0.0001 indicate significant differences. ns, no significance. Data are shown as means ± SDs, and one-way ANOVA was performed for statistical analysis.

**Figure 4 microorganisms-13-00945-f004:**
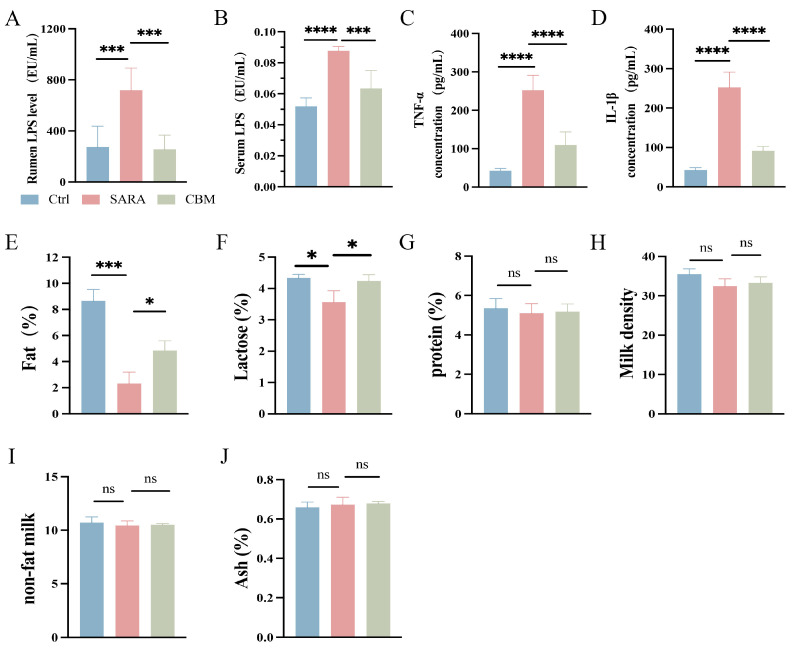
The effects of CBM on LPS and the concentration of inflammatory cytokines in SARA dairy goats. Rumen tissue, serum, and milk samples of dairy goats in Ctrl group, SARA group, and CBM group were collected for detection. (**A**,**B**) LPS content in rumen (**A**) and serum (**B**) samples of dairy goats in three different groups (n = 6). (**C**,**D**) Serum levels of inflammatory cytokines, including TNF-α (**C**) and IL-1β (**D**) concentrations, in three groups of dairy goats (n = 8). (**E**–**J**) Changes in (**E**) fat, (**F**) lactose, (**G**) protein, (**H**) density, (**I**) non-fat milk, and (**J**) ash in milk composition of three groups of dairy goats (n = 3). * *p* < 0.05, *** *p* < 0.001, and **** *p* < 0.0001 indicate significant differences. ns, no significance. Data are presented as means ± SDs, and one-way ANOVA was performed for statistical analysis.

**Figure 5 microorganisms-13-00945-f005:**
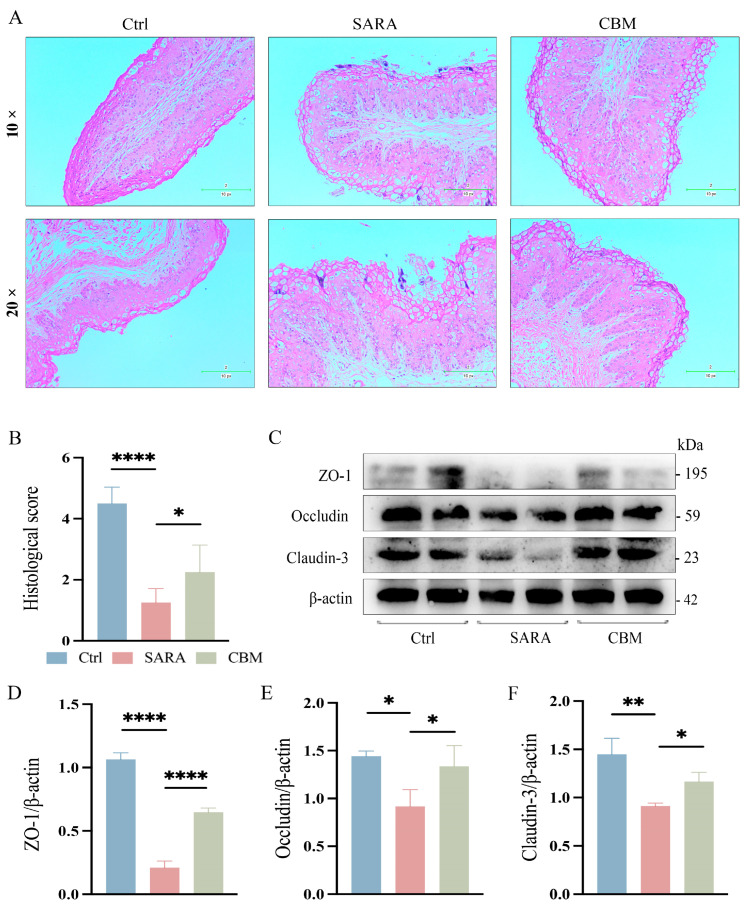
The effects of CBM on rumen barrier in SARA dairy goats. Rumen tissues of dairy goats in Ctrl group, SARA group, and CBM group were collected, and their barrier integrity was detected. (**A**) Representative H&E-stained images of rumen tissues from three different groups (100×/200×) (n = 3). (**B**) Histopathological score of the rumen tissues. (**C**–**F**) The expression levels of the TJs ZO-1, Occludin, and Claudin-3 in representative rumen tissues as detected by western blotting (**C**), and the relative intensity of the three proteins using β-actin as an internal reference (**D**–**F**) (n = 3). * *p* < 0.05, ** *p* < 0.01 and **** *p* < 0.0001 indicate significant differences. One-way ANOVA (**B**,**D**–**F**) was used, and values are presented as means ± SDs (**B**,**D**–**F**).

**Figure 6 microorganisms-13-00945-f006:**
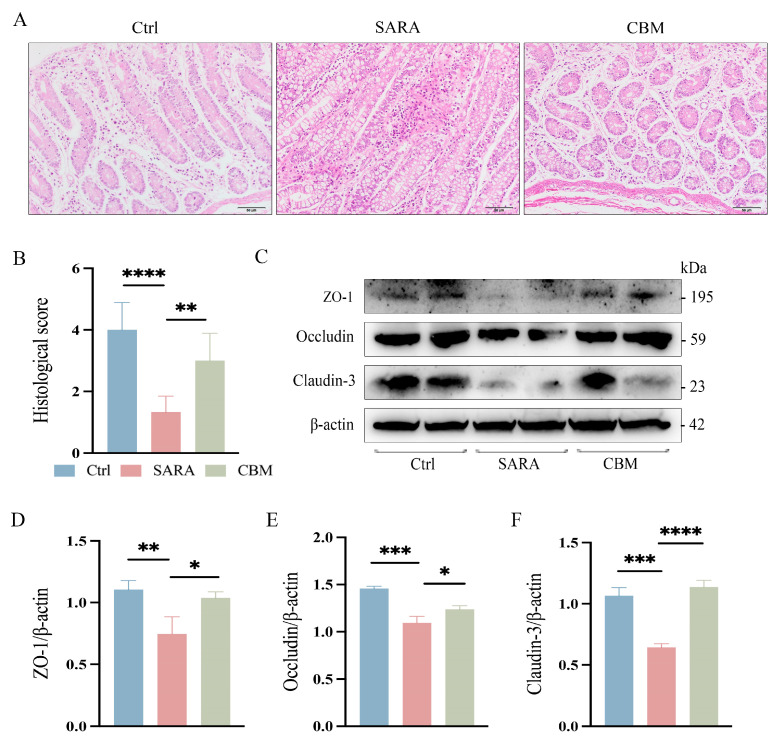
The effects of CBM on gut barrier in SARA dairy goats. Gut tissues of dairy goats in Ctrl group, SARA group, and CBM group were collected, and their barrier integrity was detected. (**A**) Representative H&E-stained images of gut tissues from three different groups (scale bar 50 μm) (n = 3). (**B**) Histopathological score of the gut tissues. (**C**–**F**) The expression levels of the TJs ZO-1, Occludin, and Claudin-3 in representative gut tissues as detected by western blotting (**C**) and the relative intensity of the three proteins using β-actin as an internal reference (**D**–**F**) (n = 3). * *p* < 0.05, ** *p* < 0.01, *** *p* < 0.001, and **** *p* < 0.0001 indicate significant differences. One-way ANOVA (**B**,**D**–**F**) was used, and values are displayed as means ± SDs (**B**,**D**–**F**).

**Figure 7 microorganisms-13-00945-f007:**
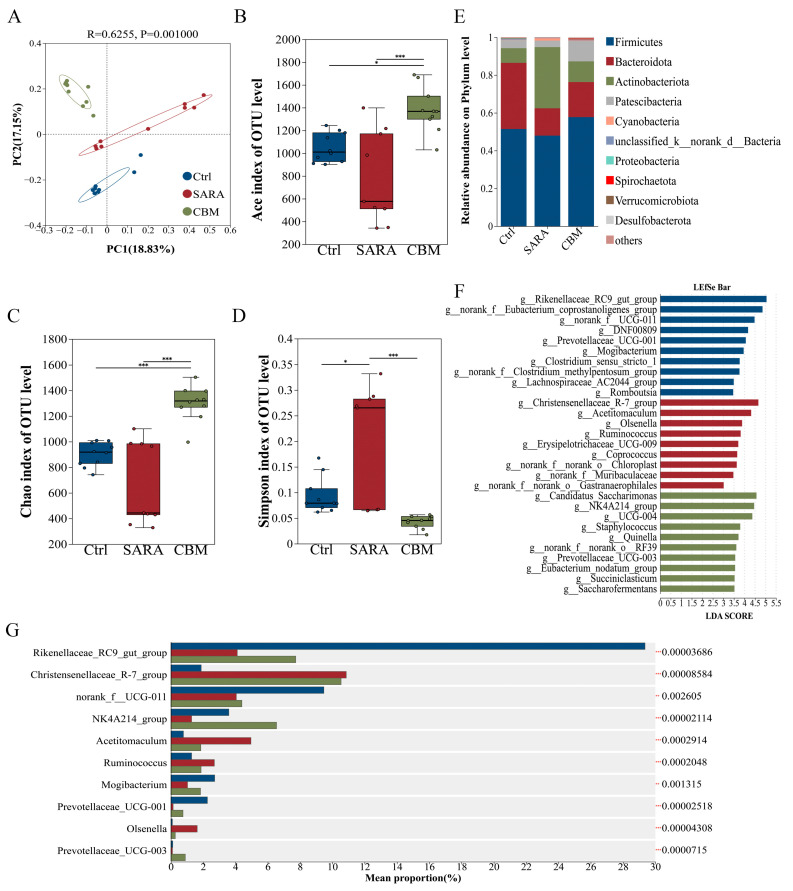
CBM treatment regulated rumen microbiota dysbiosis in SARA dairy goats. Collect rumen contents of dairy goats in Ctrl group, SARA group, and CBM group for 16S rRNA sequencing (n = 10). (**A**) The PCoA showed the structural separation of gut microbiota between the three different groups based on the unweighted Unirac distance (R = 0.6255, *p* = 0.001). (**B**–**D**) The alpha diversity index, which includes the Ace (**B**), Chao1 (**C**), and Simpson (**D**) indices. (**E**) Rumen microbial composition at the phylum level from different treatment groups. (**F**) Rumen microbial composition at the genus level from different treatment groups. (**G**) Different bacterial taxa were revealed by LEfSe to be enriched in various groups (log10 LDA score > 5.5). * *p* < 0.05 and *** *p* < 0.001 indicate significant differences.

**Figure 8 microorganisms-13-00945-f008:**
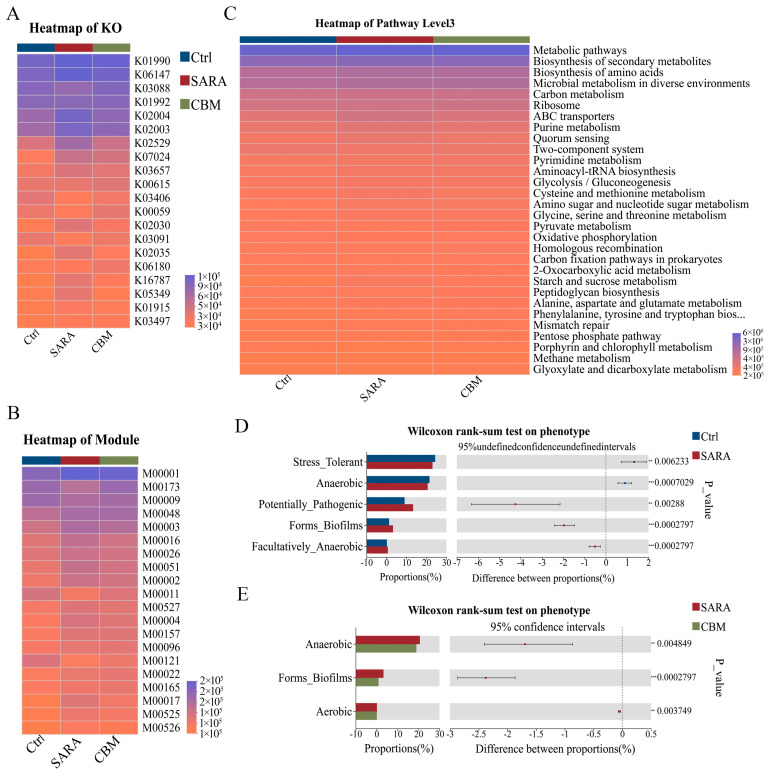
CBM treatment reversed rumen microbiota function and phenotype in SARA dairy goats. (**A**–**C**) PICRUSt was used to predict changes in the function of three groups of bacteria and was further analyzed using the Kyoto Encyclopedia of Genes and Genomes (KEGG) database. ORTHOLOGY (KO) (**A**), modules (**B**), and pathways (**C**) belonging to the KEGG functional category (**D**,**E**) based on the BugBase phenotype prediction of the three different groups.

## Data Availability

The authors confirm that the data supporting the findings of this study can be found in the manuscript. Transcriptomic data were deposited in the US National Center for Biotechnology Information (NCBI) sequence read archive under accession number PRJNA1240035.

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
