# Peer review of "Carbonate Buffer Mixture Alleviates Subacute Rumen Acidosis Induced by Long-Term High-Concentrate Feeding in Dairy Goats by Regulating Rumen Microbiota"

_microorganisms, 2025, doi:10.3390/microorganisms13040945_

Round 1

Reviewer 1 Report

Comments and Suggestions for Authors

CBM Alleviates Subacute Rumen Acidosis Induced by Long-term High-concentrate Feeding in Dairy Goats by Regulating Rumen Microbiota

In this study, SARA model was established by feeding dairy goats with a high concentrate diet (HCD) for 8 weeks, and then CBM was administered to dairy goats for two days. 

The results showed that the decrease of rumen pH caused by SARA was rapidly increased after CBM treatment. Systemic inflammation and LPS levels were reduced, and the integrity of the rumen barrier was restored.

CBM treatment up-regulated rumen microbiota abundance and diversity, and the abundance of beneficial bacteria producing butyric acids, such as Rikenellaceae_RC9_gut_group, NK4A214 group, and Prevotellaceae UCG-001 increased significantly, and the bacterial community function was restored. 

The abstract lacks experimental design, measurements, P value, and a solid conclusion

Define all abbreviations

The introduction needs citations

L33-34: use a better argument

L42: rapidly digestible starch (RDS), you should define earlier in the text

L65-69: revise and provide more information

L98-101, L112-115, L117-121: what parameters did you measure?

What were the experimental design, replication, and treatments?

Results: provide the P value in the text

Fig 7, Fig 8: fonts are too small, revise

Author Response

Comment 1: The abstract lacks experimental design, measurements, P value, and a solid conclusion.

Response 1: We gratefully appreciate for the precious time the reviewer spent making constructive remarks. We have re-written the abstract section in accordance with your request in the revised manuscript as shown below. Thanks again.

Line 12-29: This study aimed to elucidate the therapeutic mechanisms of carbonate buffer mixture (CBM) in mitigating subacute rumen acidosis (SARA) by examining its effects on rumen pH, systemic inflammation, and rumen microbiota in a dairy goat model. Using a controlled experimental design, SARA was induced through 8-week high-concentrate diet feeding (70% concentrate, 30% forage), followed by 2-day CBM treatment. Comprehensive analyses included rumen pH monitoring, serum inflammatory marker quantification (IL-1β, TNF-α) by ELISA, rumen barrier integrity assessment through tight junction protein (TJs) ZO-1, Occludin and Claudin-3 by western blot analysis, and 16S rRNA sequencing of rumen microbiota. The results demonstrated that CBM administration rapidly elevated depressed rumen pH from within 6 hours post-treatment while concurrently reducing circulating LPS levels. The analysis of rumen 16S rRNA showed that CBM significantly increased the rumen microbial diversity and abundance of SARA dairy goats. Butyric acid generation groups such as Rikenellaceae_RC9_gut_group, NK4A214 group and Prevotellaceae UCG-001 were selectively enriched, and corresponding functional predictions showed that the butyric acid synthesis pathway (PICRUSt2) was enhanced. These findings suggest that CBM has a multidimensional therapeutic effect by simultaneously correcting rumen acidosis, alleviating systemic inflammation, and restoring microbial balance through pH-dependent and pH-independent mechanisms, providing a scientifically validated nutritional strategy for SARA management in intensive ruminant production systems.

Comment 2: Define all abbreviations.

Response 2: Thanks so much for your careful check and apologize for our carelessness. We checked the full text and defined all the abbreviations in the revisions and below. Thanks again for your proposal. 

Line 33-38: The escalating global demand for dairy products has driven the widespread adoption of high-concentrate diets (HCD) in modern ruminant production systems. Scientific evidence demonstrates that HCD significantly enhances propionate-dominated rumen fermentation[1], which serves as the primary substrate for hepatic gluconeogenesis. This metabolic adaptation can significantly improve feed conversion rate (FCR) compared to traditional feed-based feeding regimens[2].

Line 47-49: However, when lactic acid production exceeds the metabolic capacity of lactic acid-utilizing bacteria, the rumen pH rapidly decreases and disturbs the rumen microbiota, inducing subacute rumen acidosis (SARA)[7].

Line 52-55: These pathological changes lead to marked reductions in feed conversion ratio (FCR) and predispose dairy ruminants to a spectrum of metabolic disturbances and infectious complications, particularly mastitis[9] and endometritis[10] that impair the health of dairy animals.

Line 92-94: Specific antibodies included beta-actin (β-actin) (#AF7018), Zonula occludens-1 (ZO-1) (#AF5145), Occludin (#DF7504) and Claudin-3 (#AF0129) were obtained from Affinity Biosciences (OH, USA).

Line 98-100: All diets meet the daily nutritional requirements needed for lactation of dairy goats and randomly allocated into three groups (n=10), the Ctrl group received a basal diet containing 30% concentrate at dry matter (DM) basis with a 7:3 forage-to-concentrate ratio.

Line 139-141: The first sample was drawn into a vacuum blood collection tube containing ethylenediaminetetraacetic acid (EDTA) as an anticoagulant for complete blood count (CBC) analysis.

Line 244-251: Blood biochemical analysis results showed that the serum concentration of aspartate aminotransferase (AST), total protein (TP), globulin (GLB), albumin (ALB), and albumin-globulin ratio (A/G) were significantly increased in SARA goats compared to the Ctrl goats. However, CBM reduced the concentration of AST, TP, GLB, and A/G compared with the SARA group (Fig. 3A-E). In addition, except for the total bile acid (TBA), CBM reduced the increases of serum total bilirubin (TB) and alkaline phosphatase (ALP) levels in SARA goats (Fig. 3F-H).

Comment 3: The introduction needs citations.

Response 3: Thanks for your careful consideration and professional comment. In response to this comment, we thoroughly examined the introduction and made appropriate citations from peer-reviewed literature to confirm all key statements. These citations provide the necessary scientific background and support for our research hypothesis. Thanks again for your valuable comments.

Comment 4: L33-34: use a better argument

Response 4: We sincerely appreciate your professional suggestions. We have re-edited this part of the content in the revised manuscript as shown below. Thanks again.

Line 33-38: The escalating global demand for dairy products has driven the widespread adoption of high-concentrate diets (HCD) in modern ruminant production systems. Scientific evidence demonstrates that HCD significantly enhances propionate-dominated rumen fermentation[1], which serves as the primary substrate for hepatic gluconeogenesis. This metabolic adaptation can significantly improve feed conversion rate (FCR) compared to traditional feed-based feeding regimens[2].

Comment 5: L42: rapidly digestible starch (RDS), you should define earlier in the text.

Response 5: Thanks for your comprehensive consideration and constructive remarks. We have checked the full text to ensure that all abbreviations are defined when they first appear. Thanks again for your comment.

Line 44-47: Meanwhile, a large amount of rapidly digestible starch (RDS) was degraded by rumen microorganisms to produce lactic acid[6], and an elevated proportion of lactic acid converted to butyric acid by lactic acid-utilizing bacteria increased[7].

Comment 6: L65-69: revise and provide more information.

Response 6: We sincerely appreciate your constructive suggestion to enhance the manuscript's completeness. We have re-edited this paragraph in the revised manuscript as shown below. Thanks again for your valuable comment.  

Line 69-76: Research has demonstrated that high-dose yeast culture supplementation (SCFPb2X) exerts beneficial effects in SARA affected dairy cows by selectively promoting cellulolytic bacteria and lactate-utilizing microorganisms while stabilizing ruminal pH and reducing free LPS concentrations[15]. In a complementary investigation, Petri et al. revealed that the combined administration of autolyzed yeast (AY) and phytogenic (PHY) through concentrate mixtures significantly modulated rumen microbiota, notably reducing the abundance of Gram-negative Succiniclasticum spp. while enhancing populations of lactate-utilizing Selenomonas spp[16].  Building upon these established findings, the present study aims to investigate whether CBM could regulate rumen microbiota while regulating rumen pH parameters. Specifically, we seek to elucidate the potential dual mechanisms through which CBM may mitigate SARA associated inflammation via direct pH stabilization and through selective microbial population shifts that could reduce LPS-producing bacterial taxa while promoting beneficial commensals. This study will provide important insights into CBM as a therapeutic mechanism for SARA in ruminants and may provide synergistic benefits through physicochemical and microbial pathways.

Comment 7: L98-101, L112-115, L117-121: what parameters did you measure?

Response 7: We sincerely appreciate for your valuable comment. We have supplemented the following details regarding the measured parameters in the revised manuscript. Thanks again.

Line 130-136: All rumen and gut tissues were fixed in 4% paraformaldehyde, embedded in paraffin, and sectioned at 5-μm. After dewaxing and hydration, sections were stained with hematoxylin and eosin (H&E). Histopathological evaluation was performed under an Olympus microscope (Tokyo, Japan) to assess severity of epithelial injury (from absent to mild), and the extent of inflammatory cell in filtrate (from none or rare to transmural). A semi-quantitative scoring system adapted from[8] was applied to grade the severity of each parameter.

Line 146-151: The dairy goat teat area was sterilized using 75% ethanol and the first three handfuls of milk were discarded to minimize contamination. Approximately 50 mL of milk was then aseptically collected into a sterile tube containing potassium dichromate as a preservative.  Milk composition analysis was performed using a Milkoscan™ FT1 infrared analyzer (FOSS Analytical A/S, Hillerød, Denmark) to determine fat, protein, lactose, milk density, non-fat milk, and ash content.

Line 153-163: The collected rumen tissue was mixed with PBS in a 1:9 ratio to produce a 10% tissue homogenate. Centrifuge 10 minutes at 4 °C, 12,000× g, collect supernatant for detection. Blood samples are centrifuged under the same conditions to collect the upper serum. The proinflammatory cytokines TNF-α and IL-1β were measured in tissue supernatants and serum samples using commercially available ELISA kits, following the manufacturer’s protocols. Briefly, 100 μL of standards or diluted samples (1:5 in assay buffer) were added to antibody-precoated wells and incubated for 2 hours at 37 °C. After washing five times with 0.5% TBST, 100 μL biotinylated detection antibody was added (1 hour, 37 °C), followed by HRP-streptavidin (30 minutes) and TMB substrate (15 minutes). The reaction was stopped with Hâ‚‚SOâ‚„, and absorbance was measured at 450 nm (reference 570 nm).

Comment 8: What were the experimental design, replication, and treatments?

Response 8: We sincerely appreciate the reviewer's insightful comment regarding the need for clearer experimental design documentation. Our study utilized a completely randomized design with three parallel treatment groups: (1) healthy ctrl goats fed a 30% concentrate diet (forage: concentrate ratio of 7:3), (2) SARA model goats gradually adapted to 70% concentrate over 8 weeks, and (3) CBM-treated SARA goats receiving daily oral administration of CBM for 2 days. The experimental design incorporated ten biological replicates per group, with triplicate technical replicates for all analytical measurements. The complete protocol included a 2-week acclimation period followed by 8 weeks of SARA induction, 2 days of treatment administration, and subsequent sample collection, with all procedures approved by the Jilin University Institutional Animal Care and Use Committee. The revision part has been highlighted in the revised manuscript, and the modification examples are as follows. Thanks again.

Line 96-119: Thirty dairy goats (2-3 years, average weight of 55 kg) were obtained from a farm in Changchun, Jilin Province, China, and no diseased animals and no history of antibiotic therapy. All diets meet the daily nutritional requirements needed for lactation of dairy goats and randomly allocated into three groups (n=10), the Ctrl group received a basal diet containing 30% concentrate at dry matter (DM) basis with a 7:3 forage-to-concentrate ratio; the SARA model group underwent gradual adaptation where the concentrate proportion increased from 30% to 70% over 21 days to induce SARA, maintaining the final 7:3 ratio for 8 weeks; and the CBM treatment group received oral administration of a CBM mixture, which is mainly composed of Na2CO3 (25 g), NaHCO3 (200 g), NaCl (50 g), MgSO4 (25 g), CaCl2 (2.5 g), KCl (10 g) dissolved in 5 L of water, dosing of SARA-affected goat, each goat was dosed once a day for a total of 2 days. The experimental diets comprised a series of nutritionally balanced formulations with increasing concentrate levels from 30%to 70% of DM. The formulations systematically replaced alpha hay (decreasing from 56% to 26% DM) with higher proportions of corn meal (32-56% DM) and soybean meal (8-14% DM), while maintaining constant inclusion rates of limestone (2% DM), salt (1% DM), and a vitamin-mineral premix (1% DM). This premix provided comprehensive micronutrient supplementation, containing per kilogram: 450 mg nicotinic acid, 600 mg Mn, 950 mg Zn, 430 mg Fe, 650 mg Cu, 30 mg Se, 45 mg I, 20 mg Co, 800 mg vitamin E, 45,000 IU vitamin D, and 120,000 IU vitamin A. The nutritional profiles demonstrated progressive increases in metabolic energy (8.19-18.34 MJ/kg DM) and starch content (17.12-37.95% DM), while maintaining consistent crude protein levels (15% DM) across all diets. Corresponding reductions in fiber fractions were observed, with neutral detergent fiber decreasing from 52% to 25% DM and acid detergent fiber from 35% to 15% DM.

Line 272-260: Figure 4. The effects of CBM on LPS and the concentration of inflammatory cytokines in SARA dairy goats.Rumen tissue, serum, and milk samples of dairy goats in Ctrl group, SARA group and CBM group were collected for detection. A-B. LPS content in rumen (A) and serum (B) samples of dairy goats in three different groups (n=6). C-D. Serum levels of inflammatory cytokines, including TNF-α (C) and IL-1β (D) concentrations , in three groups of dairy goats (n=8). E-J. Changes of E. Fat. F. Lactose. G. Protein. H. Density. I. non-fat milk. J. Ash. in milk composition of three groups of dairy goats (n=3). * p < 0.05, ** p < 0.01, *** p < 0.001 and **** p < 0.0001 indicate significant differences. ns, no significance. Data are presented as means ± SDs and one-way ANOVA was performed for statistical analysis.

Comment 9: Results: provide the P value in the text.

Response 9: Thanks for your constructive comment. We appreciate your suggestion to enhance statistical transparency in the Results section. In response, we have revised the manuscript to report p-values for all significant comparisons, while retaining asterisk notations in figures for clarity.  For non-significant trends, we now provide the precise p-values instead of labeling them as ‘ns’ (not significant), enabling readers to evaluate effect sizes more comprehensively. Thanks again.

Line 282-291: When LPS translocases into the peripheral blood circulation and causes metabolic changes in the body, nutrients are redistributed and a large amount of nutrients are used for the body's immune response, thus reducing the amount of nutrients going to the mammary glands for the synthesis of milk composition, reduced quality of the milk. In the SARA group, we found that the content of milk fat and lactose decreased significantly (p<0.001 and p <0.05), but the synthesis rate of milk fat and lactose increased after CBM treatment (both p<0.05; Fig. 4D-E). In SARA model, milk protein and milk density tended to decrease, but their contents did not change after CBM treatment (p=0.9733 and p=0.8017; Fig. 4F-H). In addition, non-fat milk, ash, and other milk compositions did not change in the two groups (p=0.9773 and p=0.9522; Fig. 4I-J).

Comment 10: Fig 7, Fig 8: fonts are too small, revise

Response 10: We appreciate your careful examination and suggestions. We have adjusted the font in Fig.7 and Fig.8 in revised manuscript. Thanks again for your constructive comment.

Reviewer 2 Report

Comments and Suggestions for Authors

microorganisms-3555364-peer-review-v1

This is interesting work showing the effects if treatment of acidosis and consequences for the biological markers and microbial populations in goats.

The results of the present study indicate that CBM treatment can alleviate SARA by improving rumen microbiota, reducing rumen epithelial inflammatory cell infiltration, and reducing LPS content in the rumen and circulation. Furthermore, authors pointed out that CBM treatment resulted in a notable alteration of the rumen microbiota composition, with a significant up-regulate in the relative abundance of beneficial bacteria, including Rikenellaceae_RC9_gut_group, NK4A214 group, and Prevotellaceae UCG-001. In conclusion, the results of this study confirm that CBM can alleviate SARA-induced rumen inflammation and systemic inflammation by regulating the composition of the rumen microbiota.

Introduction is informative and provide a basis for the introducing readers into the project research plan.

Please, centrifugation needs to be as g force, not rpm (see Ln 105, 109, 118, etc)

Please, provide more details regarding tests under section 2.7. Elisa

Ln137: In this and similar occasion, please, change to USA.

For all suppliers of material and equipment, please, following the recommendations form the Publisher and Journal. When the supplier was mentioned for the first time, the name of the company and full address needed to be provided, including city, state (in case of federal country) in an abbreviated way, and name of the country. In following occasions, only name of the company will be sufficient. Example: BioRad, Hercules, CA, USA.

Please, always try to use the address of the headquarters of the company and not local distributors.

Results are presented in an appropriate way, however, in some occasions a bit more detail for the applied procedures will be more than welcome.

Please, be sure that all results claimed in the paper are associated with appropriately described material and methods.

Discussion need to be extended. In the current manuscript very detailed description of the results was provided, however, discussion section is very telegraphic and in fact sound much more as short conclusion that discussion of the obtained results and observations.

Concussion can be elaborated a bit more. Authors will need to state clear take home messages with principal observations and pave an ideas for the potential further investigations.

References needs to be formatted according to instructions from publisher and the journal.

Author Response

Comment 1: Please, centrifugation needs to be as g force, not rpm (see Ln 105, 109, 118, etc)

Response 1: We gratefully appreciate for your constructive comments. As suggested, we have now revised all centrifugation steps in the manuscript to explicitly report the relative centrifugal force (× g) instead of rpm. All modifications have been highlighted in the revised manuscript as shown below. Thanks again.

Line 141-144: The second sample was collected in a sterile, pyrogen-free microcentrifuge tube and subsequently processed by centrifugation at 1000× g for 10 minutes at 4 °C.

Line 153-155: The collected rumen tissue was mixed with PBS in a 1:9 ratio to produce a 10% tissue homogenate. Centrifuge 10 minutes at 4 °C, 12,000× g, collect supernatant for detection.

Line 165-166: After weighing 0.03 g rumen tissue, 270 μL RIPA lysate was added for grinding, centrifugation at 12,000× g and 4 °C for 10 minutes to collect supernatant.

Comment 2: Please, provide more details regarding tests under section 2.7. Elisa.

Response 2: We thank you for your rigorous consideration and constructive comments. Detailed ELISA protocols have been added to Section 2.7 in revised manuscript as follows. Thanks again.

Line 153-163: The collected rumen tissue was mixed with PBS in a 1:9 ratio to produce a 10% tissue homogenate. Centrifuge 10 minutes at 4 °C, 12,000× g, collect supernatant for detection. Blood samples are centrifuged under the same conditions to collect the upper serum. The proinflammatory cytokines TNF-α and IL-1β were measured in tissue supernatants and serum samples using commercially available ELISA kits, following the manufacturer’s protocols. Briefly, 100 μL of standards or diluted samples (1:5 in assay buffer) were added to antibody-precoated wells and incubated for 2 hours at 37 °C. After washing five times with 0.5% TBST, 100 μL biotinylated detection antibody was added (1 hour, 37 °C), followed by HRP-streptavidin (30 minutes) and TMB substrate (15 minutes). The reaction was stopped with Hâ‚‚SOâ‚„, and absorbance was measured at 450 nm (reference 570 nm). 

Comment 3: Ln137: In this and similar occasion, please, change to USA.

Response 3: Thanks so much for your valuable comment. We have revised the country name to "USA" throughout the manuscript as suggested and shown below. Thanks again.

Line 178-180: Total bacterial DNA extraction Microbial community genomic DNA was extracted from rumen samples of dairy goats using the E.Z.N.A.® soil DNA kit (Omega Bio-tek, Norcross, GA, USA) according to manufacturer's instructions.

Comment 4: For all suppliers of material and equipment, please, following the recommendations form the Publisher and Journal. When the supplier was mentioned for the first time, the name of the company and full address needed to be provided, including city, state (in case of federal country) in an abbreviated way, and name of the country. In following occasions, only name of the company will be sufficient. Example: BioRad, Hercules, CA, USA.

Response 4: We have carefully revised all supplier information throughout the manuscript to comply with the journal's formatting requirements. For initial mentions of commercial sources, we now provide full company names with complete addresses, while subsequent references retain only the company name. This adjustment has been uniformly applied to reagents, equipment , and software in both the main text and supplementary materials. Thanks again for your comment.

Line 87-94: The reagents including Na2CO3, Na2HCO3, NaCl, MgSO4, CaCl2, KCl was obtained from Tianli Chemical Reagent Co., Ltd (Tianjin, China). The LPS test kit (cat.YX-121618G) was obtained from Horseshoe Crab Reagent Manufactory Co., Ltd. (Xiamen, China). Tumor necrosis factor (TNF)-α (cat.430915) and interleukin (IL)-1β (cat.432615) enzyme-linked immunosorbent assay (ELISA) kits were obtained from BioLegend (San Diego, CA, USA). Specific antibodies included beta-actin (β-actin) (#AF7018), Zonula occludens-1 (ZO-1) (#AF5145), Occludin (#DF7504) and Claudin-3 (#AF0129) were obtained from Affinity Biosciences (Cincinnati, OH, USA).

Line 130-132: Histopathological evaluation was performed under a microscope (Olympus Corporation, Tokyo, Japan) to assess severity of epithelial injury (from absent to mild), and the extent of inflammatory cell in filtrate (from none or rare to transmural).

Line 149-151: Milk composition analysis was performed using a Milkoscan™ FT1 infrared analyzer (FOSS Analytical A/S, Hillerød, Denmark) to determine fat, protein, lactose, milk density, non-fat milk, and ash content.

Line 178-180: Total bacterial DNA extraction Microbial community genomic DNA was extracted from rumen samples of dairy goats using the E.Z.N.A.® soil DNA kit (Omega Bio-tek, Norcross, GA, USA) according to manufacturer's instructions.

Line 180-185: After testing the DNA quality with 2% agarose gel, the DNA concentration and purity were determined using NanoDrop2000 (Thermo Fisher Scientific, Waltham, MA, USA). Using the ABI GeneAmp® 9700 PCR thermal circulator (Applied Biosystems, Foster City, CA, USA), bacterial 16S rRNA genes V3-V4 were amplified with 338F (5'-ACT CCT ACG GGA GGC AGC AG-3') and 806R (5'-GGA CTA CHVGGG TWT CTAAT-3').

Line 188-191: Products of PCR were recovered with 2% agarose gel, purified according to the instructions of DNA Gel Recovery and Purification Kit (MajorBio, Shanghai, China), and quantified with Qubit 4.0 (Thermo Fisher Scientific).

Line 191-193: The products were sequenced using the Illumina PE300/PE250 platform (Illumina, San Diego, CA, USA) according to the standard protocol of Majorbio.

Line 203-204: All data in this study were analyzed and plotted using GraphPad Prism 8.0 (GraphPad Software, San Diego, CA, USA).

Comment 5: Please, always try to use the address of the headquarters of the company and not local distributors.

Response 5: Thanks so much for your valuable comments. Throughout the revised manuscript, we have now systematically revised all supplier addresses to exclusively use the corporate headquarters information. Thanks again for your remarks.

Comment 6: Results are presented in an appropriate way, however, in some occasions a bit more detail for the applied procedures will be more than welcome.

Response 6: Thanks so much for your valuable and constructive suggestions. We sincerely appreciate the reviewer's constructive feedback regarding the need for additional methodological details in the Results section. In response, we have carefully revised the manuscript to provide more comprehensive descriptions of the experimental procedures underlying key findings.     Specifically, we have expanded the explanation of statistical analyses to clarify the exact tests used (e.g., one-way ANOVA with Tukey’s post-hoc test), included critical technical parameters (e.g., sample sizes for each experiment). These modifications ensure full transparency and reproducibility while maintaining the conciseness of the Results section. All additions are highlighted in the revised manuscript for the reviewer’s convenience, example as shown below. Thanks again.

Line 238-242: Figure 2. The effects of CBM on Blood-RT in SARA dairy goats. Blood samples of dairy goats in Ctrl group, SARA group and CBM group were collected for CBC detection. Changes of A. WBC. B. Neu%. C. Lym%. D. RBC. E. HGB. F. PLT. in blood samples of three different groups of dairy goats (n=3). *p < 0.05, ** p < 0.01, *** p < 0.001 and **** p < 0.0001 indicate significant differences. Data are displayed as means ± SDs and one-way ANOVA was performed for statistical analysis.

Comment 7: Please, be sure that all results claimed in the paper are associated with appropriately described material and methods.

Response 7: We sincerely appreciate this critical comment. We have carefully addressed this issue by examining manuscripts to ensure that all reported results are fully consistent with their corresponding methodological descriptions. We have expanded the Materials and Methods section to include the precise technical details of each analysis process, such as experimental design, detection indicators, etc. We have confirmed that the methods corresponding to all experimental results mentioned in the results are fully described, including algorithm parameters and significance thresholds. Thanks again.

Comment 8: Discussion need to be extended. In the current manuscript very detailed description of the results was provided, however, discussion section is very telegraphic and in fact sound much more as short conclusion that discussion of the obtained results and observations.

Response 8: Thank you for your constructive suggestion. In response, we completely rewrote the discussion section in revised manuscript. We provide an in-depth analysis of the potential mechanisms by which CBM affects microbial population dynamics and physicochemical parameters in the rumen environment and strengthen the manuscript's contribution to this field as shown below. Thanks again for your comment.

Line 389-450: The development of SARA in ruminants is predominantly associated with the sustained administration of high-concentrate rations to enhance FCR. It is characterized by prolonged rumen pH below 5.8 and rumen microbiota disturbance. When SARA occurs in dairy goats, it causes metabolic disorders such as diarrhea, decreased dry matter intake, weight loss and decreased milk production, resulting in huge economic losses to the dairy industry[11]. The treatment of SARA can be in various ways, such as increasing the proportion of roughage in the diet to promote saliva secretion[17-19], adding plant extracts to enhance fermentation, and incorporating probiotics to inhibit lactobacillus. Solution of CBM is a basic solution composed of inorganic compounds such as Na2CO3, NaHCO3, MgSO4, NaCl, MgSO4, CaCl2, KCl. It can neutralize excess free acid in the rumen, which can quickly relieve the decrease in rumen pH. The aim was to investigate whether CBM could alleviate SARA and possible regulatory mechanisms.

Ruminants SARA was diagnosed when rumen pH remained <5.8 for ≥3 h/day[20]. After 8 weeks of HCD feeding, the SARA model was successfully established and then treated with CBM for 2 days. Results indicate that CBM effectively alleviated SARA-induced rumen pH dysregulation by chemically buffering excess volatile fatty acids (VFAs) and free H+ ions. In contrast, NaHCO₃ enhanced rumen buffering capacity over time by stimulating salivary secretion and modulating rumen osmotic pressure in SARA-affected goats[21]. The decline in rumen pH promoted the proliferation and lysis of Gram-negative bacteria, releasing substantial amounts of LPS. Upon binding to TLR4, LPS activated the NF-κB/MAPK signaling pathway, triggering the excessive production of pro-inflammatory cytokines (e.g., TNF-α, IL-1β)[22]. Consistent with this mechanism, SARA-affected dairy goats in our study exhibited a marked increase in systemic inflammation. Simultaneously, excessive LPS reduced milk fat percentage in SARA-affected animals by decreasing ruminal acetic acid (a key milk fat precursor) and inhibiting critical mammary lipogenic enzymes, including acetyl-CoA carboxylase and fatty acid synthetase[23]. Furthermore, LPS upregulated matrix metalloproteinases (MMPs)[24], degraded TJs, compromised rumen barrier integrity, translocated to mammary tissue, and ultimately impaired mammary epithelial cell function, leading to metabolic dysfunction, reduced glucose availability, and decreased lactose synthesis[25]. These findings were consistently observed in our experimental results. Notably, CBM treatment effectively mitigated these adverse effects by significantly increasing rumen pH in SARA-affected dairy goats. Previous studies have demonstrated that exogenous additives can modulate rumen microbiota to reduce LPS levels, thereby alleviating SARA[15,16]. Building on this evidence, we hypothesized that CBM could further optimize rumen microbial composition by suppressing Gram-negative bacteria proliferation and consequently limiting LPS production. To test this hypothesis, we investigated the regulatory effects of CBM on rumen microbiota. Sequencing analysis of 16S rRNA revealed that CBM treatment effectively restored SARA-induced alterations in rumen microbial diversity and abundance. At the phylum level, CBM administration significantly increased Firmicutes abundance, a Gram-positive bacteria phylum, thereby competitively suppressing Gram-negative bacterial proliferation. As the primary nutrient source for gastrointestinal epithelial cells, butyrate levels demonstrate a positive correlation with gastrointestinal permeability[26]. In our study, the enriched Firmicutes phylum was identified as the predominant functional bacterial group responsible for butyrate synthesis, with notable producers including Lachnospiraceae, Ruminococcaceae, Eubacteriaceae, and Fusobacteriaceae[27-29]. Butyrate has been well-documented to possess anti-inflammatory properties, offering protection against gastrointestinal pathogens through maintenance of bacterial enzyme activity. Simultaneously, the abundances of Bacteroidetes and Verrucomicrobiota were significantly elevated in the CBM-treated group. Bacteroidetes contribute to rumen epithelial energy metabolism through SCFAs production, while Verrucomicrobiota enhance barrier function by modulating mucus layer thickness[30,31]. The study further demonstrated that CBM treatment significantly increased Actinobacteria abundance, which exhibited a negative correlation with inflammatory markers (WBC and Neu%). Additionally, CBM administration markedly enhanced the abundance of beneficial microbiota including Rikenellaceae_RC9_gut_group, NK4A214_group, and Prevotellaceae_UCG-001. These microbiota-derived metabolites modulate rumen epithelial cell function, proliferation, and secretory activity through multiple signaling pathways, thereby influencing rumen motility, and enhancing barrier integrity[32-34]. Furthermore, this microbial-metabolic crosstalk impacts host metabolic homeostasis and ameliorates both local and systemic inflammatory responses.

Comment 9: Conclusion can be elaborated a bit more. Authors will need to state clear take home messages with principal observations and pave an ideas for the potential further investigations.

Response 9: Thanks so much for your thorough review. We have re-edited the discussion section in revised manuscript as shown below. Thanks again.

Line 452-460: This study substantiates our initial hypothesis that CBM exerts therapeutic effects on SARA through integrated physicochemical and microbial regulatory mechanisms.  The findings demonstrate CBM's capacity to address the research objectives by simultaneously modulating rumen pH parameters and microbial community structure, thereby attenuating both local and systemic inflammatory responses. These results advance our understanding of nutritional interventions for SARA by establishing CBM's dual-action mechanism that reconciles microbial ecology with rumen physiological homeostasis. This evidence supports the potential application of CBM in the treatment of SARA in ruminant and sheds light on its mechanisms.

Comment 10: References needs to be formatted according to instructions from publisher and the journal.

Response 10: We are sorry of our careless and sincerely appreciate this critical reminder. Through the following measures, all references have been fully reformatted to comply with the MDPI reference style as specified in the Microorganisms journal guidelines (2024 edition) as shown in the following example. Thanks again.

Line 33-36: The escalating global demand for dairy products has driven the widespread adoption of high-concentrate diets (HCD) in modern ruminant production systems. Scientific evidence demonstrates that HCD significantly enhances propionate-dominated rumen fermentation[1].
1.    Orton, T.; Rohn, K.; Breves, G.; Brede, M. Alterations in fermentation parameters during and after induction of a subacute rumen acidosis in the rumen simulation technique. J Anim Physiol Anim Nutr (Berl) 2020, 104, 1678-1689, doi:10.1111/jpn.13412.

Reviewer 3 Report

Comments and Suggestions for Authors

The title should not begin with acronyms. This is not considered scientific writing.

Likewise, sentences should not begin with acronyms. Correct this throughout the text of the article.

All acronyms should be defined before their use; check this throughout the text.

The abstract should begin with the objective, followed by a brief description of what was evaluated, presentation of the results, and final considerations. Seek to improve this item.

Remove the words contained in the title from the keywords

Line 51 Numerous studies.... The authors cite only 2 studies. Rewrite the sentence or increase the number of study citations.

Lines 65-69 - This information should be included in the research method.

Insert a hypothesis and an objective at the end of the introduction

What ingredients made up the concentrate? What diet was offered? What Roughage? A table with the ingredients of the diet and the diet offered should be included. The nutritional composition of the ingredients and the diet must also be included.

What is the experimental design and statistical model used?

The resolution of the figures must be improved

The conclusion must be reformulated. At this stage, it is not necessary to present the results, but rather a response to the objectives and the hypothesis tested in this research.

Comments on the Quality of English Language

The title should not begin with acronyms. This is not considered scientific writing.

Likewise, sentences should not begin with acronyms. Correct this throughout the text of the article.

All acronyms should be defined before their use; check this throughout the text.

The abstract should begin with the objective, followed by a brief description of what was evaluated, presentation of the results, and final considerations. Seek to improve this item.

Remove the words contained in the title from the keywords

Line 51 Numerous studies.... The authors cite only 2 studies. Rewrite the sentence or increase the number of study citations.

Lines 65-69 - This information should be included in the research method.

Insert a hypothesis and an objective at the end of the introduction

What ingredients made up the concentrate? What diet was offered? What Roughage? A table with the ingredients of the diet and the diet offered should be included. The nutritional composition of the ingredients and the diet must also be included.

What is the experimental design and statistical model used?

The resolution of the figures must be improved

The conclusion must be reformulated. At this stage, it is not necessary to present the results, but rather a response to the objectives and the hypothesis tested in this research.

Author Response

Reviewer #3:

Comment 1: The title should not begin with acronyms. This is not considered scientific writing.

Response 1: We gratefully appreciate for your positive remarks and valuable comments. As suggested, we have revised the title to avoid starting with an acronym. The modified title now reads: "Carbonate buffer mixture Alleviates Subacute Rumen Acidosis Induced by Long-Term High-Concentrate Feeding in Dairy Goats by Regulating Rumen Microbiota". Thanks again for your time and constructive comment. 

Comment 2: Likewise, sentences should not begin with acronyms. Correct this throughout the text of the article.

Response 2: We sincerely appreciate this constructive comment. Throughout the revised manuscript, we have systematically corrected all instances where sentences began with acronyms as shown below. Thanks again.

Line 50-54: The impaired rumen environment characteristic of SARA significantly diminishes fiber fermentation efficiency while inducing structural damage to the rumen epithelium. These pathological changes lead to marked reductions in feed conversion ratio (FCR) and predispose dairy ruminants to a spectrum of metabolic disturbances and infectious complications, particularly mastitis[8] and endometritis[9] that impair the health of dairy animals.

Line 87-88: The reagents including Na2CO3, Na2HCO3, NaCl, MgSO4, CaCl2, KCl was obtained from Tianli Chemical Reagent Co., Ltd, Tianjin, China.

Line 88-89: The LPS test kit (cat.YX-121618G) was obtained from Chinese Horseshoe Crab Reagent Manufactory Co., Ltd., Xiamen, China.

Line 188-190: Products of PCR were recovered with 2% agarose gel, purified according to the instructions of DNA Gel Recovery and Purification Kit (MajorBio, China), and quantified with Qubit 4.0 (Thermo Fisher Scientific, USA).

Line 196-201: To characterize intergroup variations in microbial composition, LDA effect size (LEfSe) analysis was employed to detect phylogenetically discriminative features with statistical significance. Functional potential of the microbiota was subsequently predicted through phylogenetic investigation of communities by reconstruction of unobserved states (PICRUSt2) based on 16S rRNA gene sequencing data.

Line 364-367: Functional prediction of rumen microbiota under CBM treatment was performed using PICRUSt. The predicted functional profiles were subsequently annotated and an-alyzed through the Kyoto Encyclopedia of Genes and Genomes (KEGG) database to elucidate potential metabolic pathways.

Line 389-390: The development of SARA in ruminants is predominantly associated with the sustained administration of high-concentrate rations to enhance FCR.

Line 394-397: The treatment of SARA can be in various ways, such as increasing the proportion of roughage in the diet to promote saliva secretion[15-17], adding plant extracts to enhance fermentation, and incorporating probiotics to inhibit lactobacillus.

Line 397-398: Solution of CBM is a basic solution composed of inorganic compounds such as Na2CO3, Na2HCO3, NaCl, MgSO4, CaCl2, KCl.

Comment 3: All acronyms should be defined before their use; check this throughout the text.

Response 3: We gratefully appreciate your professional advice. We checked the full text and defined all the abbreviations in the revisions and below. Thanks again for your proposal. 

Line 12-14: This study aimed to elucidate the therapeutic mechanisms of carbonate buffer mixture (CBM) in mitigating subacute rumen acidosis (SARA) by examining its effects on rumen pH, systemic inflammation, and rumen microbiota in a dairy goat model.

Line 16-19: Comprehensive analyses included rumen pH monitoring, serum inflammatory marker quantification (IL-1β, TNF-α) by ELISA, rumen barrier integrity assessment through tight junction protein (TJs) ZO-1, Occludin and Claudin-3 by western blot analysis, and 16S rRNA sequencing of rumen microbiota.

Line 33-38: The escalating global demand for dairy products has driven the widespread adoption of high-concentrate diets (HCD) in modern ruminant production systems. Scientific evidence demonstrates that HCD significantly enhances propionate-dominated rumen fermentation[1], which serves as the primary substrate for hepatic gluconeogenesis. This metabolic adaptation can significantly improve feed conversion rate (FCR) compared to traditional feed-based feeding regimens[2].

Line 41-44: This leads to a decrease in the amount of bicarbonate and dihydrogen phosphate entering the rumen and a decrease in the ability of the rumen to neutralize organic acids. Short-chain fatty acids (SCFAs) are absorbed by the rumen wall at a lower rate than produced, causes the pH of the rumen begins to decrease[3-5].

Line 47-49: However, when lactic acid production exceeds the metabolic capacity of lactic acid-utilizing bacteria, the rumen pH rapidly decreases and disturbs the rumen microbiota, inducing subacute rumen acidosis (SARA).

Line 50-54: The impaired rumen environment characteristic of SARA significantly diminishes fiber fermentation efficiency while inducing structural damage to the rumen epithelium. These pathological changes lead to marked reductions in feed conversion ratio (FCR) and predispose dairy ruminants to a spectrum of metabolic disturbances and infectious complications, particularly mastitis[9] and endometritis[10] that impair the health of dairy animals.

Line 64-67: Some studies have found that after SARA occurs in sheep, due to the death and lysis of Gram-negative bacteria such as Bacteroides and the release of endotoxin (LPS), the relative abundance of Bacteroidetes decreases while the concentration of rumen LPS increases significantly[15].

Line 72-76: In a complementary investigation, Petri et al. revealed that the combined administration of autolyzed yeast (AY) and phytogenic (PHY) through concentrate mixtures significantly modulated rumen microbiota, notably reducing the abundance of Gram-negative Succiniclasticum spp. while enhancing populations of lactate-utilizing Selenomonas spp[17].

Line 90-94: Tumor necrosis factor (TNF)-α (cat.430915) and interleukin (IL)-1β (cat.432615) enzyme-linked immunosorbent assay (ELISA) kits were obtained from BioLegend (San Diego, CA, USA). Specific antibodies included beta-actin (β-actin) (#AF7018), Zonula occludens-1 (ZO-1) (#AF5145), Occludin (#DF7504) and Claudin-3 (#AF0129) were obtained from Affinity Biosciences (Cincinnati, OH, USA).

Line 98-100: All diets meet the daily nutritional requirements needed for lactation of dairy goats and randomly allocated into three groups (n=10), the Ctrl group received a basal diet containing 30% concentrate at dry matter (DM) basis with a 7:3 forage-to-concentrate ratio.

Line 138-141: Blood samples was collected from the jugular vein of goats using a standardized protocol. The first sample was drawn into a vacuum blood collection tube containing ethylenediaminetetraacetic acid (EDTA) as an anticoagulant for complete blood count (CBC) analysis.

Line 244-251: Blood Biochemical analysis results showed that the serum concentration of aspartate aminotransferase (AST), total protein (TP), globulin (GLB), albumin (ALB), and albumin-globulin ratio (A/G) were significantly increased in SARA goats compared to the Ctrl goats. However, CBM reduced the concentration of AST, TP, GLB, and A/G compared with the SARA group (Fig. 3A-E). In addition, except for the total bile acid (TBA), CBM reduced the increases of serum total bilirubin (TB) and alkaline phosphatase (ALP) levels in SARA goats (Fig. 3F-H).

Comment 4: The abstract should begin with the objective, followed by a brief description of what

was evaluated, presentation of the results, and final considerations. Seek to improve this item.

Response 4: Thank you so much for your valuable comment. We have rewritten the abstract according to the linear structure (IMRaD format) of “goal-method-result-conclusion” in revised manuscript as shown below. Thanks again for your constructive suggestions.

Line 12-29: This study aimed to elucidate the therapeutic mechanisms of carbonate buffer mixture (CBM) in mitigating subacute rumen acidosis (SARA) by examining its effects on rumen pH, systemic inflammation, and rumen microbiota in a dairy goat model. Using a controlled experimental design, SARA was induced through 8-week high-concentrate diet feeding (70% concentrate, 30% forage), followed by 2-day CBM treatment. Comprehensive analyses included rumen pH monitoring, serum inflammatory marker quantification (IL-1β, TNF-α) by ELISA, rumen barrier integrity assessment through tight junction protein(TJs) ZO-1, Occludin and Claudin-3 by western blot analysis, and 16S rRNA sequencing of rumen microbiota. The results demonstrated that CBM administration rapidly elevated depressed rumen pH from within 6 hours post-treatment while concurrently reducing circulating LPS levels. The analysis of rumen 16S rRNA showed that CBM significantly increased the rumen microbial diversity and abundance of SARA dairy goats. Butyric acid generation groups such as Rikenellaceae_RC9_gut_group, NK4A214 group and Prevotellaceae UCG-001 were selectively enriched, and corresponding functional predictions showed that the butyric acid synthesis pathway (PICRUSt2) was enhanced. These findings suggest that CBM has a multidimensional therapeutic effect by simultaneously correcting rumen acidosis, alleviating systemic inflammation, and restoring microbial balance through PH-dependent and PH-independent mechanisms, providing a scientifically validated nutritional strategy for SARA management in intensive ruminant production systems.

Comment 5: Remove the words contained in the title from the keywords.

Response 5: We strongly agree with your comments and thank you for your suggestions. We have revised the keywords section to eliminate any terms that duplicate words already present in the manuscript title, while maintaining the most relevant and distinctive terms that accurately represent the study's focus. Thanks again.

Line 30: Keywords: Rumen pH; LPS; Inflammation ; Rumen Barrier Permeability

Comment 6: Line 51 Numerous studies.... The authors cite only 2 studies. Rewrite the sentence or increase the number of study citations.

Response 6: Thank you for your careful consideration. We have already rephrased the sentence as shown below. Thanks again for your constructive comment.

Line 56-57: Studies have shown that rumen microorganisms contain the largest proportion of bacteria and are highly sensitive to changes in rumen fluid pH[11,12].

Comment 7: Lines 65-69 - This information should be included in the research method.

Response 7: Thank you for your constructive comment. We have moved this part to the methods section in the revised manuscript as shown below. Thanks again for your reminder.

Line 96-106: All diets meet the daily nutritional requirements needed for lactation of dairy goats and randomly allocated into three groups (n=10), the Ctrl group received a basal diet containing 30% concentrate at dry matter (DM) basis with a 7:3 forage-to-concentrate ratio; the SARA model group underwent gradual adaptation where the concentrate proportion increased from 30% to 70% over 21 days to induce SARA, maintaining the final 7:3 ratio for 8 weeks; and the CBM treatment group received oral administration of a CBM mixture, which is mainly composed of Na2CO3 (25 g), NaHCO3 (200 g), NaCl (50 g), MgSO4 (25 g), CaCl2 (2.5 g), KCl (10 g) dissolved in 5 L of water, dosing of SARA-affected dairy goat, each dairy goat was dosed once a day for a total of 2 days.

Comment 8: Insert a hypothesis and an objective at the end of the introduction.

Response 8: We sincerely appreciate your constructive suggestion to strengthen the conceptual framework of our study. We have carefully revised the concluding paragraph of the introduction to explicitly state both the central hypothesis and specific research objectives. Thanks again.

Line 77-84: Building upon these established findings, the present study aims to investigate whether CBM could regulate rumen microbiota while regulating rumen pH parameters. Specifically, we seek to elucidate the potential dual mechanisms through which CBM may mitigate SARA-associated inflammation via direct pH stabilization and through selective microbial population shifts that could reduce LPS-producing bacterial taxa while promoting beneficial commensals. This study will provide important insights into CBM as a therapeutic mechanism for SARA in ruminants and may provide synergistic benefits through physicochemical and microbial pathways.

Comment 9: What ingredients made up the concentrate? What diet was offered? What Roughage? A table with the ingredients of the diet and the diet offered should be included. The nutritional

composition of the ingredients and the diet must also be included.

Response 9: Thanks so much for your valuable suggestions. We have supplemented the detailed composition of the diet in the Materials and Methods section of the revised draft as shown below. Thanks again.

Line 107-119: The experimental diets comprised a series of nutritionally balanced formulations with increasing concentrate levels from 30% to 70% of DM. The formulations systematically replaced alpha hay (decreasing from 56% to 26% DM) with higher proportions of corn meal (32-56% DM) and soybean meal (8-14% DM), while maintaining constant inclusion rates of limestone (2% DM), salt (1% DM), and a vitamin-mineral premix (1% DM). This premix provided comprehensive micronutrient supplementation, containing per kilogram: 450 mg nicotinic acid, 600 mg Mn, 950 mg Zn, 430 mg Fe, 650 mg Cu, 30 mg Se, 45 mg I, 20 mg Co, 800 mg vitamin E, 45,000 IU vitamin D, and 120,000 IU vitamin A. The nutritional profiles demonstrated progressive increases in metabolic energy (8.19-18.34 MJ/kg DM) and starch content (17.12-37.95% DM), while maintaining consistent crude protein levels (15% DM) across all diets. Corresponding reductions in fiber fractions were observed, with neutral detergent fiber decreasing from 52% to 25% DM and acid detergent fiber from 35% to 15% DM.

Comment 10: What is the experimental design and statistical model used?

Response 10: We appreciate for your professional comment. This study employed a completely randomized block design with three experimental groups (Ctrl, SARA model, and CBM treatment;   n=10 goats/group). The statistical model for continuous outcomes like cytokine levels used one-way ANOVA with Tukey’s post-hoc test, while count data like microbial OTUs were analyzed via generalized linear mixed models (GLMM) with negative binomial distribution. Thanks again for your valuable comment.

Line 96-119: Thirty dairy goats (2-3 years, average weight of 55 kg) were obtained from a farm in Changchun, Jilin Province, China, and no diseased animals and no history of antibiotic therapy. All diets meet the daily nutritional requirements needed for lactation of dairy goats and randomly allocated into three groups (n=10), the Ctrl group received a basal diet containing 30% concentrate at dry matter (DM) basis with a 7:3 forage-to-concentrate ratio; the SARA model group underwent gradual adaptation where the concentrate proportion increased from 30% to 70% over 21 days to induce SARA, maintaining the final 7:3 ratio for 8 weeks; and the CBM treatment group received oral administration of a CBM mixture, which is mainly composed of Na2CO3 (25 g), NaHCO3 (200 g), NaCl (50 g), MgSO4 (25 g), CaCl2 (2.5 g), KCl (10 g) dissolved in 5 L of water, dosing of SARA-affected goat, each goat was dosed once a day for a total of 2 days. The experimental diets comprised a series of nutritionally balanced formulations with increasing concentrate levels from 30%to 70% of DM. The formulations systematically replaced alpha hay (decreasing from 56% to 26% DM) with higher proportions of corn meal (32-56% DM) and soybean meal (8-14% DM), while maintaining constant inclusion rates of limestone (2% DM), salt (1% DM), and a vitamin-mineral premix (1% DM). This premix provided comprehensive micronutrient supplementation, containing per kilogram: 450 mg nicotinic acid, 600 mg Mn, 950 mg Zn, 430 mg Fe, 650 mg Cu, 30 mg Se, 45 mg I, 20 mg Co, 800 mg vitamin E, 45,000 IU vitamin D, and 120,000 IU vitamin A. The nutritional profiles demonstrated progressive increases in metabolic energy (8.19-18.34 MJ/kg DM) and starch content (17.12-37.95% DM), while maintaining consistent crude protein levels (15% DM) across all diets. Corresponding reductions in fiber fractions were observed, with neutral detergent fiber decreasing from 52% to 25% DM and acid detergent fiber from 35% to 15% DM.

Line 238-242: Figure 2. The effects of CBM on Blood-RT in SARA dairy goats. Blood samples of dairy goats in Ctrl group, SARA group and CBM group were collected for CBC detection. Changes of A. WBC. B. Neu%. C. Lym%. D. RBC. E. HGB. F. PLT. in blood samples of three different groups of dairy goats (n=3). *p < 0.05, ** p < 0.01, *** p < 0.001 and **** p < 0.0001 indicate significant differences. Data are displayed as means ± SDs and one-way ANOVA was performed for statistical analysis.

Comment 11: The resolution of the figures must be improved.

Response 11: We sincerely appreciate your valuable comment. We have systematically improved all figures by regenerating them at higher resolution (minimum 600 dpi) using vector-based formats for line graphs and 300 dpi TIFF format for micrographs. We have verified that all text elements remain legible at 100% magnification and that color contrasts meet accessibility standards. These modifications have been implemented throughout the manuscript to enhance the visual representation of our findings. Thanks again for your constructive comment.

Comment 12: The conclusion must be reformulated. At this stage, it is not necessary to present the results, but rather a response to the objectives and the hypothesis tested in this research.

Response 12: Thanks for your valuable suggestion. We have re-edited the discussion section in revised manuscript as shown below. Thanks again for your comment.

Line 452-460: This study substantiates our initial hypothesis that CBM exerts therapeutic effects on SARA through integrated physicochemical and microbial regulatory mechanisms.  The findings demonstrate CBM's capacity to address the research objectives by simultaneously modulating rumen pH parameters and microbial community structure, thereby attenuating both local and systemic inflammatory responses. These results advance our understanding of nutritional interventions for SARA by establishing CBM's dual-action mechanism that reconciles microbial ecology with rumen physiological homeostasis. This evidence supports the potential application of CBM in the treatment of SARA in ruminant and sheds light on its mechanisms.

Round 2

Reviewer 1 Report

Comments and Suggestions for Authors

Carbonate Buffer Mixture Alleviates Subacute Rumen Acidosis Induced by Long-Term High-Concentrate Feeding in Dairy Goats by Regulating Rumen Microbiota.

Thank you for providing a revised version of the manuscript.

Author Response

Thank you for reviewing the manuscript.

Reviewer 2 Report

Comments and Suggestions for Authors

Authors have improved the quality of the manuscript, with specific focus on material and methods and Discussion sections. In my opinion paper can be suggested to be accepted for publication. 

Author Response

Thank you for reviewing the manuscript.